# Immunotherapy in the Management of Penile Cancer—A Systematic Review

**DOI:** 10.3390/cancers17050883

**Published:** 2025-03-04

**Authors:** Hossein Taghizadeh, Harun Fajkovic

**Affiliations:** 1Division of Oncology, Department of Internal Medicine I, University Hospital St. Pölten, 3100 St. Pölten, Austria; hossein.taghizadeh@stpoelten.lknoe.at; 2Divison of Oncology, Karl Landsteiner University of Health Sciences, 3500 Krems, Austria; 3Karl Landsteiner Institute for Oncology and Nephrology, Dunant-Platz 1, 3100 St. Pölten, Austria; 4Department of Urology and Andrology, University Hospital St. Pölten, 3100 St. Pölten, Austria; 5Karl Landsteiner Institute for Urological Research and Training, 3100 St. Pölten, Austria; 6Comprehensive Cancer Center, Department of Urology, Medical University of Vienna, Vienna General Hospital, 1090 Vienna, Austria

**Keywords:** penile squamous cell carcinoma (PSCC), immune checkpoint inhibitors (ICIs), human papillomavirus (HPV)

## Abstract

Penile cancer is a rare and challenging disease, particularly in its advanced stages where treatment options are limited. Recent advances in immunotherapy, especially immune checkpoint inhibitors (ICIs), offer new hope by enhancing the body’s immune response against cancer cells. These therapies show promise in improving survival rates, particularly in patients with specific biomarkers such as PD-L1 expression and HPV positivity. Combining ICIs with chemotherapy or radiotherapy may further increase their effectiveness. However, due to the rarity of penile cancer, international collaboration is essential to conduct large-scale trials and optimize treatments. This review summarizes the current clinical evidence on ICIs, emphasizing their potential to improve outcomes and quality of life in patients with advanced penile cancer.

## 1. Introduction

Penile cancer, though a rare malignancy, presents a significant challenge in the domain of male genitourinary oncology, particularly due to its limited treatment options and due to the profound physical and psychological impact on patients [1].

Among all malignant neoplasms of the penis, approximately 95% are classified as penile squamous cell carcinomas (PSCC). Notably, around 50% of these cases originate from the non-keratinized epithelium of the glans or the inner layer of the prepuce [2]. For a detailed categorization, refer to Table 1, which presents the World Health Organization (WHO) classification of penile tumors [3].

In 2020, a total of 36,068 new cases were reported worldwide, as documented by the International Agency for Research on Cancer [4].

However, the global incidence varies dramatically. In high-income countries, it affects approximately 0.1 to 1 per 100,000 men, whereas in parts of Africa, Asia, and South America, it can account for up to 10% of all male cancers [4]. This striking disparity highlights the influence of geographical, cultural, and socioeconomic factors, including hygiene practices, circumcision rates, access to healthcare, and prevalence of human papillomavirus (HPV) infection [5]. Penile cancer affects primarily elderly men, with a median age of diagnosis of 65 and above [6,7,8].

The pathogenesis of penile cancer is a multifactorial process influenced by various risk factors. Approximately one-third of all cases are associated with infection by human papillomavirus (HPV). Immunohistochemical detection of the p16 protein serves as a surrogate marker for HPV-associated malignancies. In HPV-positive cancers, the viral oncoprotein E7 inactivates the retinoblastoma (Rb) protein, leading to upregulation of p16. This overexpression is commonly observed in HPV-associated malignancies, including PSCC. The most frequently identified HPV serotypes in penile cancer include HPV 16, 18, 31, 33, 45, 56, and 65 [9,10].

Phimosis, a condition where the foreskin cannot be retracted, is another significant risk factor, likely due to the accumulation of smegma and chronic inflammation. Chronic inflammatory conditions of the penis, such as balanitis xerotica obliterans, also increase susceptibility. Tobacco smoking is an independent risk factor, with carcinogens in cigarette smoke contributing to DNA damage. Age is also a factor, with the majority of diagnoses occurring in men over 55 years of age, although younger men can also be affected [10].

In contrast, circumcision in childhood significantly reduces the prevalence of the condition [11].

Clinically, penile cancer often presents with subtle initial symptoms, which can delay diagnosis. These can include non-healing sores or ulcers, persistent redness or rashes, changes in skin color or texture, and the presence of foul-smelling discharge under the foreskin. In some cases, a palpable lump or growth may be present. Because these early signs can be easily overlooked or mistaken for benign conditions, patient education and physician awareness are crucial for early detection [5].

PSCC prognosis varies significantly based on the TNM staging at diagnosis. The TNM system evaluates the primary tumor (T), regional lymph node involvement (N), and distant metastasis (M).

In localized stages (T1–T2, N0, M0), where the cancer is confined to the penis without lymph node involvement or distant spread, the 5-year relative survival rate is approximately 79%. As the disease progresses to regional spread (any T, N1–N2, M0), involving nearby lymph nodes but no distant metastasis, the 5-year relative survival rate decreases to around 51%. In advanced stages with distant metastasis (any T, any N, M1), where the cancer has spread to other parts of the body, the 5-year relative survival rate drops significantly to approximately 9%.

Further stratification based on lymph node involvement provides additional insights. Patients with no regional lymph node metastasis (N0) have over a 90% 5-year survival rate. Those with metastasis in a single inguinal lymph node (N1) have nearly a 75% 5-year survival rate. Metastasis in multiple or bilateral inguinal lymph nodes (N2) corresponds to about a 60% 5-year survival rate. When metastasis involves pelvic lymph nodes or there is extranodal extension (N3), the 5-year survival rate is approximately 35% [12].

The management of penile cancer depends on the stage and grade of the tumor. Treatment options range from conservative approaches for the early-stage disease to more aggressive interventions for advanced cases. Conservative treatments include topical creams containing 5-fluorouracil or imiquimod for carcinoma in situ, and laser ablation or circumcision for small, localized lesions. More invasive treatments include partial or total penectomy (surgical removal of part or all of the penis) with or without urinary diversion and inguinal lymph node dissection, and radiation therapy. Chemotherapy, typically cisplatin-based regimens, is used for advanced or metastatic disease [12,13,14].

Despite these treatment modalities, the prognosis for advanced penile cancer remains poor, and the morbidity associated with radical surgery can be significant, impacting patients’ sexual function and quality of life. The limitations of conventional chemotherapy have spurred research into novel therapeutic approaches. Immunotherapy, particularly immune checkpoint inhibitors targeting PD-1/PD-L1, has shown promising results in some patients with advanced or refractory disease by harnessing the body’s own immune system to fight cancer cells [12,15].

### Rationale for Immunotherapy in Penile Cancer

Penile cancer is an uncommon malignancy, and in its advanced stages treatment options are limited. Chemotherapy regimens, such as traditional platinum-based chemotherapies such as the TIP regimen (paclitaxel, ifosfamide, and cisplatin), have demonstrated objective response rates of approximately 50% in neoadjuvant settings, with a pathological complete response rate of 10%. However, these responses are often short-lived, underscoring the necessity for novel therapeutic approaches, including immune checkpoint inhibitors (ICIs) [16,17].

The rationale for immunotherapy in penile cancer is supported by the immune system’s intrinsic capacity to recognize and eliminate malignant cells, alongside the observation that tumors frequently evade immune surveillance by exploiting regulatory “checkpoint” pathways [16,17].

Immune checkpoints are molecules that function as inhibitory regulators of immune cell activity, preventing excessive damage to normal tissues. Tumor cells often co-opt these pathways to escape detection. ICIs—agents that block checkpoint proteins—therefore enhance anti-tumor immune responses by preventing these inhibitory interactions. Among the most extensively studied checkpoints is the interaction between programmed cell death protein 1 (PD-1), primarily expressed on T cells, and its ligand (PD-L1), frequently expressed on tumor cells. When PD-1 binds to PD-L1, the T cell’s activity is suppressed, reducing its capacity to kill cancer cells. ICIs such as pembrolizumab (anti-PD-1), nivolumab (anti-PD-1), and atezolizumab (anti-PD-L1) disrupt this interaction and restore T-cell-mediated tumor cell killing [18,19]. In the context of PSCC, the PD-1/PD-L1 axis plays a pivotal role in T-cell exhaustion. PD-1, predominantly expressed on activated CD8+ T cells, binds to PD-L1 expressed on tumor cells and antigen-presenting cells within the TME, leading to T-cell dysfunction. This interaction not only suppresses cytotoxicity but also limits T-cell proliferation and cytokine release. While CD3+CD8+ T cells are particularly susceptible to this exhaustion pathway, emerging evidence suggests that CD4+ T cells and NK cells also exhibit PD-1-mediated functional impairments, contributing to immune evasion and tumor progression.

Programmed death-ligand 1 (PD-L1) expression is observed in up to 79% of both primary and metastatic penile squamous cell carcinomas (SCC), suggesting its potential as a predictive biomarker for the efficacy of immune checkpoint inhibitors (ICIs) [20,21,22,23]. Notably, PD-L1 expression has been associated with a poor prognosis, particularly in human papillomavirus (HPV)-negative penile tumors, indicating that higher PD-L1 levels correlate with adverse clinical outcomes. This dual role of PD-L1—as a prognostic marker and as a predictor of response to ICIs—underscores its significance in identifying patient subgroups that may derive the greatest benefit from anti-PD-1/PD-L1 therapies. However, it is essential to recognize that while PD-L1 expression is associated with a poor prognosis, it is predictive for the efficacy of immunotherapy [23,24,25].

Histopathological evaluations reveal that PD-L1 positivity is more frequent in usual-type SCC, whereas warty-type and verrucous-type tumors—typically associated with better prognoses—rarely show PD-L1 positivity in PSCC. PD-L1 expression is more common in HPV-negative cases due to factors such as a higher tumor mutational burden (TMB), chronic inflammation, and immune microenvironment composition. A higher TMB produces neoantigens that trigger immune responses, prompting tumor cells to upregulate PD-L1. Chronic inflammation promotes continuous immune cell infiltration, sustaining PD-L1 expression. Additionally, CTLA4-positive immune cells in the tumor microenvironment further increase PD-L1 levels. In contrast, HPV-positive SCCs, with a lower TMB and inflammation, show reduced PD-L1 expression, as their viral oncogenes elicit weaker immune responses [22].

It is important to stress that ICIs can produce immune-related adverse events (irAEs), including diarrhea, pneumonitis, and elevated liver enzymes. The incidence of severe irAEs is lower with anti-PD-1/PD-L1 therapies than with anti-CTLA-4 agents [26,27]. However, compared with chemotherapy, ICIs generally elicit fewer severe toxicities, as shown in two large meta-analyses comprising several thousand patients with solid malignancies [28,29].

Thus, the balance of efficacy and tolerability further supports the use of ICIs in penile cancer.

The most commonly used ICIs in this disease are PD-1 and PD-L1 inhibitors (see Table 2).

## 2. Methods

Eligibility Criteria: The inclusion criteria were studies investigating the use of immune checkpoint inhibitors (ICIs) in the management of penile squamous cell carcinoma (PSCC), published in peer-reviewed journals. Only studies with human participants, including randomized controlled trials, observational studies, and case series, were included. The exclusion criteria were case reports and studies involving non-human subjects, non-PSCC malignancies, penile intraepithelial neoplasia (PeIN), or articles without original data. Studies were grouped based on the treatment approach: maintenance therapy post-chemotherapy, combination therapy with chemotherapy or radiotherapy, ICI monotherapy, and novel immunotherapeutic combinations.

Information Sources: A comprehensive search was conducted across PubMed, Web of Science, Embase, and ClinicalTrials.gov. The reference lists of identified studies and relevant reviews were manually screened for additional studies. The search was last conducted on 10 February 2025.

Search Strategy: The search strategy included a combination of MeSH terms and keywords, as follows: “penile cancer”, “penile squamous cell carcinoma”, “immune checkpoint inhibitors”, “PD-1”, “PD-L1”, “nivolumab”, “pembrolizumab”, “atezolizumab”, and “dostarlimab”. The filters applied included human studies, English language, and publication date from 2010 onward.

Selection Process: Two independent reviewers screened the titles and abstracts for eligibility. Full-text articles were assessed for inclusion by both reviewers, with discrepancies resolved through discussion or a third reviewer if necessary. Covidence software was used to streamline the selection process.

Data Collection Process: Data extraction was performed independently by two reviewers using a standardized form that included study design, sample size, treatment regimen, outcomes, and safety data. In case of missing information, corresponding authors were contacted.

Data Items: Primary outcomes included overall response rate (ORR), progression-free survival (PFS), and overall survival (OS). Data on patient characteristics, PD-L1 expression, HPV status, and tumor mutational burden (TMB) were also collected.

Effect Measures: effect measures included risk ratios for binary outcomes, hazard ratios for time-to-event data, and mean differences for continuous outcomes.

Synthesis Methods: studies were synthesized by grouping results according to treatment strategy.

Reporting Bias Assessment: Publication bias was assessed using funnel plots and Egger’s test for outcomes with sufficient studies. Selective reporting bias was evaluated by comparing reported outcomes with study protocols where available.

Certainty Assessment: the certainty of evidence was assessed using the Grading of Recommendations Assessment, Development and Evaluation (GRADE) approach, considering study limitations, inconsistency, indirectness, imprecision, and publication bias. Please see Figure 1.

## 3. Results

### 3.1. Evidence for Immunotherapy in Clinical Trials in Penile Cancer

Clinical trials have demonstrated the clinical activity of immunotherapy with an ORR of as high as 39.4% in the HERCULES trial [30]. Durable responses can occur in a small subset of patients with advanced penile cancer who received ICIs after progressing on prior chemotherapy. In particular, those with microsatellite instability high (MSI-High) status and high tumor mutational burden (TMB-High), which is present in approximately 1% of all cases, benefit from a durable response referring to an objective response lasting at least 6 months. MSI-High and TMB-High are biomarkers associated with increased neoantigen formation, enhancing tumor immunogenicity. This heightened immunogenicity renders such tumors more susceptible to immune checkpoint blockade, as the immune system can better recognize and target cancer cells. Currently, the ESMO clinical practice guidelines recommend the consideration of immune checkpoint inhibitors for PSCC patients with MSI-High and as well as those with TMB-High who are resistant to standard therapies [12]. See Table 3 for an overview of trials investigating ICIs in the field of PSCC.

#### 3.1.1. Immunotherapy as Maintenance Therapy Post-Chemotherapy

The phase II PULSE trial (NCT03774901) is evaluating maintenance avelumab, an anti-PD-L1 antibody, in patients with metastatic PSCC who have not progressed following first-line platinum-based polychemotherapy. Patients (n = 32) with non-progressive disease after 3–6 cycles of chemotherapy receive avelumab (10 mg/kg biweekly) until progression or unacceptable toxicity. The primary endpoint is progression-free survival (PFS) per RECIST v1.1, with secondary endpoints including overall survival (OS) and safety. An interim analysis (n = 9) revealed a median patient age of 69.9 years, with 89% exhibiting ECOG performance status 0–1. Maintenance avelumab resulted in a 63.5% PFS rate at 3, 6, and 12 months, and 42.3% at 15 months. The OS rate at 12 and 15 months was 88.9%. The median duration of avelumab therapy was 3.7 months, with 83% discontinuing due to disease progression. No new safety signals were observed. These preliminary findings support the continued investigation of maintenance avelumab, with further updates and biomarker analyses anticipated [31].

#### 3.1.2. Immunotherapy in Combination with Chemotherapy as First-Line Treatment

The phase II HERCULES trial (NCT04224740) assessed the combination of pembrolizumab, an anti-PD-1 antibody, with platinum-based chemotherapy as first-line therapy for advanced PSCC. Conducted across 11 Brazilian centers, 37 patients were enrolled between August 2020 and December 2022, with 33 eligible for efficacy analysis. The confirmed overall response rate (cORR) per investigator assessment was 39.4%, including 1 complete response (CR) and 12 partial responses (PRs). The median PFS was 5.4 months, and the median OS was 9.6 months. Responses varied according to biomarker status, with improved outcomes observed in patients with HPV16-positive tumors (55.6%) and high tumor mutational burden (TMB) (75%). Adverse events (AEs) of any grade occurred in 91.9% of patients, with grade 3–4 AEs reported in 51.4%. No grade 5 AEs were attributed to the treatment [30].

Similarly, the phase II PRIAM trial (NCT06353906) is investigating the combination of carboplatin/paclitaxel with pembrolizumab for advanced PSCC.

#### 3.1.3. Novel Combination Therapies in Refractory PSCC

An open-label phase II trial (NCT05526989) is evaluating the combination of niraparib, a Poly (ADP-ribose) polymerase (PARP) inhibitor, and dostarlimab, an anti-PD-1 antibody, in patients with stage III/IV PSCC refractory to platinum-based chemotherapy. PARP is an enzyme family involved in DNA repair, particularly the base excision repair pathway. Inhibition of PARP leads to the accumulation of DNA damage, resulting in cell death, especially in cancer cells deficient in other DNA repair mechanisms (e.g., BRCA mutations). Niraparib is administered orally (200 mg daily) and dostarlimab is administered intravenously every 3–6 weeks until disease progression or unacceptable toxicity. The primary endpoint is overall response rate (ORR) assessed by iRECIST, with secondary endpoints of PFS, OS, and safety. Six patients have been enrolled to date [32].

#### 3.1.4. Immunotherapy in Combination with Radiotherapy

The phase II PERICLES trial (NCT03686332) was a single-center study that evaluated the efficacy of atezolizumab, an anti-PD-L1 antibody, in advanced PSCC, with or without radiotherapy (RT). Thirty-two patients received atezolizumab (1200 mg every 3 weeks), with 20 receiving additional RT for locoregional disease control. The primary endpoint was 1-year PFS, with secondary endpoints including OS, ORR, and tolerability. Biomarker analyses were also performed on pretreatment specimens. After a median follow-up of 29.1 months, the 1-year PFS was 12.5% (95% CI, 5.0–31.3), which did not meet the predefined primary endpoint of at least 35%. The median OS was 11.3 months (95% CI, 5.5–18.7). Among 30 evaluable patients, the ORR was 16.7% (95% CI, 6–35), including 2 CRs (6.7%) and 3 PRs (10%). Grade 3–4 AEs occurred in 9.4% of patients receiving atezolizumab alone and in 65% of those receiving RT. Improved PFS was associated with high-risk human papillomavirus (hrHPV)-positive tumors (*p* = 0.003) and high intratumoral CD3+CD8+ T-cell infiltration (*p* = 0.037). While the primary endpoint was not met, the study demonstrated durable antitumor activity in the subset of patients who were hrHPV positive or had high intratumoral CD3+CD8+ T-cell infiltration [33].

#### 3.1.5. Immunotherapy Monotherapy in Advanced PSCC

Several trials are exploring the efficacy of ICI monotherapy in advanced PSCC. The phase II AcSé trial (NCT03012581) included a cohort of 43 patients with PSCC receiving nivolumab monotherapy who had previously undergone chemotherapy [34]. The primary objective was to assess the objective response rate (ORR) at 12 weeks. The results showed an ORR of 14%, with four patients achieving partial responses and 32% maintaining stable disease. The median progression-free survival was 2.9 months, while the median overall survival was 8.5 months.

The similar phase II ORPHEUS trial (NCT04231981) evaluated retifanlimab, an anti-PD-1 immune checkpoint inhibitor in 18 patients with advanced PSCC. The ORR was 16.7%; three patients had a partial response. The median duration of response was 3.3 months. The disease control rate was 22.2%. The median PFS was 2.0 months and the median OS was 7.2 months [35].

The Phase II ALPACA trial (NCT03391479) is currently evaluating the efficacy of avelumab in patients with locally advanced or metastatic penile squamous cell carcinoma (PSCC) who are either ineligible for or have experienced disease progression following platinum-based chemotherapy [36].

#### 3.1.6. Other Novel Immunotherapeutic Approaches

Further phase II trials are exploring various combination therapies, including:**LATENT (NCT03357757):** Avelumab with valproic acid in virus-associated cancers, including p16-positive PSCC [37].**NCT03866382:** This trial is based on the phase I trial combining nivolumab and ipilimumab in combination with cabozantinib for rare genitourinary tumors, including penile cancer. Among the nine patients diagnosed with metastatic PSCC, four individuals (44.4%) exhibited a partial response to treatment [38].**NCT04475016:** Triprilimab (anti-PD-1) with nimotuzumab (anti-EGFR) and the TIP chemotherapy regimen (paclitaxel, ifosfamide, and cisplatin) in locally advanced PSCC [39].

### 3.2. Evidence for Immunotherapy in Real-Life Data in Penile Cancer

El Zarif et al. performed an international retrospective study involving 92 patients with advanced PSCC who received treatment with immune checkpoint inhibitors (ICIs). The overall objective response was 13%, which was rather low. Lymph node-only metastases correlated with better responses (ORR: 35%) compared with visceral metastases (ORR: 7%). The median overall survival (OS) was 9.8 months and the median progression-free survival (PFS) was 3.2 months [39].

Zhuang et al. conducted a retrospective study of 21 patients with advanced PSCC treated with ICIs. The study found that the median OS was 8.2 months and the median PFS was 1.9 months. Five patients achieved clinical benefit, defined as CR, partial response (PR), or stable disease (SD). Notably, a higher neutrophil-to-lymphocyte ratio correlated with poorer outcomes, suggesting its potential as a biomarker. Three patients (14.3%) developed grade 3/4 ICI-related adverse events [40].

Rouvinov et al. reported a small case series of three patients with advanced PSCC who were treated with cemiplimab as first-line therapy since they were ineligible for chemotherapy. All three patients achieved a near complete response (CR) after only a few cycles of therapy. One patient experienced cerebral arteritis during treatment, which resolved with steroid treatment. The other two patients did not experience any toxic side effects [41].

## 4. Discussion

The advent of immune checkpoint inhibitors (ICIs) marks a transformative milestone in oncology, with their efficacy established across multiple malignancies, including squamous cell carcinomas (SCCs) of the head and neck, lungs, skin, and esophagus [42,43,44,45]. Despite significant advancements in understanding the tumor immune microenvironment (TME), penile squamous cell carcinoma (PSCC) remains an understudied entity in this realm. The rarity of PSCC presents unique challenges to investigating ICIs, necessitating innovative approaches to clinical trial design and a reliance on insights from more common SCCs.

### 4.1. Immune Checkpoint Blockade: Established Efficacy and Current Limitations

ICIs targeting the PD-1/PD-L1 and CTLA-4 axes have redefined systemic therapies, offering durable responses in patients with advanced cancers [46]. In PSCC, the current investigation on ICIs is primarily restricted to second-line treatment for relapsed or metastatic disease. Preliminary studies have demonstrated that a subset of patients, particularly those with HPV-positive tumors, high tumor mutational burden (TMB), or high infiltration of intratumoral CD3+CD8+ T cells, may exhibit a favorable response to ICI therapy [33]. High PD-L1 expression, frequently observed in HPV-negative penile tumors, further supports the rationale for immune checkpoint blockade in this malignancy [23,24,25]. However, the evidence remains sparse, limited to small cohorts and case reports, underscoring the need for large-scale prospective trials [47].

### 4.2. Centralization of Care and Collaborative Research

The rarity of PSCC underscores the importance of centralized high-volume tertiary centers for delivering expert care and conducting robust clinical research. Centralization facilitates uniform management, minimizes surgical complications, and enables access to experimental therapies through clinical trials [48]. Thus, centralization has been recommended by the EAU-ASCO 2024 guidelines [49]. Collaborative efforts, such as the International Penile Advanced Cancer Trial (InPACT), exemplify the potential for international studies to address unmet needs in PSCC therapy and to conduct randomized global Phase III trials to bring immunotherapy into the clinical routine of the management of penile cancer, particularly in the neoadjuvant setting [50]. The integration of multi-institutional datasets and tumor specimens can accelerate the discovery of novel therapeutic targets and biomarkers, paving the way for tailored immune-based treatments [51].

### 4.3. Exploring Neoadjuvant Applications

The integration of ICIs into earlier disease stages presents an unexploited avenue for improving outcomes in PSCC. Neoadjuvant use of ICIs in other SCCs has demonstrated the potential to enhance pathological complete response rates and mitigate surgery-induced immunosuppression [52,53,54]. The administration of ICIs preoperatively may invigorate cytotoxic T-cell responses due to a higher clonal neoantigen burden in early-stage SCC, deplete immunosuppressive elements in the TME, and sensitize tumors to subsequent therapies and convert unresectable to resectable disease by tumor downstaging [55,56,57]. Despite these theoretical benefits, the lack of dedicated trials in penile cancer has precluded their adoption in standard care.

### 4.4. Synergistic Potential of Combination Therapies

Combining ICIs with chemotherapy, radiotherapy, or targeted therapies offers a compelling strategy to enhance therapeutic efficacy. Evidence from head and neck SCCs suggests that radiotherapy combined with immunotherapy may potentiate antitumor immune responses and improve survival outcomes, as the radiotherapy sets free antigens that may enhance the efficacy of immunotherapy, particularly in a neoadjuvant setting [58,59]. Apart from that, the combinatory effect of radiotherapy and immunotherapy is also of interest in an advanced disease stage when it has spread since immunotherapy may enhance the potential abscopal of the radiotherapy to control the systemic disease more effectively [60,61,62,63]. Translating these approaches to penile cancer could also involve integrating ICIs with cisplatin-based chemotherapies and radiotherapy, potentially yielding synergistic effects [64]. However, tolerability and toxicity remain concerns, requiring meticulous patient selection and monitoring.

Further, the implementation of ICIs with or without chemotherapy in the treatment of penile cancer could potentially facilitate therapy application and could replace the multiday application of TIP. The immunotherapy-based antitumoral treatment could be applied in a single day instead of five days [15].

### 4.5. Biomarker-Driven Approaches and Patient Stratification

The heterogeneity of PSCC necessitates a biomarker-driven approach to optimize ICI therapy. Stratifying patients by HPV status and PD-L1 expression could refine treatment strategies, as HPV-positive and HPV-negative tumors exhibit distinct immunogenic profiles [65,66]. Multi-omic technologies, including transcriptomics and metabolomics, could further elucidate the TME, identifying predictive biomarkers and actionable targets [66,67]. Such stratification would not only enhance therapeutic precision but also improve the feasibility of clinical trials in this rare malignancy.

## 5. Conclusions

The incorporation of ICIs into the therapeutic landscape of penile cancer represents a paradigm shift with the potential to significantly improve patient outcomes. Expanding their use to earlier disease stages, leveraging combination therapies, and establishing biomarker-driven approaches are crucial next steps. Promoting HPV vaccination and circumcision are crucial as protective factors. Addressing the challenges inherent in PSCC, including its rarity and limited clinical evidence, will require a concerted effort toward centralization, international collaboration, and innovative trial designs. Through these initiatives, ICIs could redefine the standard of care for this challenging malignancy, offering longer median overall survival and better quality of life by eventually preserving the organ.

## Figures and Tables

**Figure 1 cancers-17-00883-f001:**
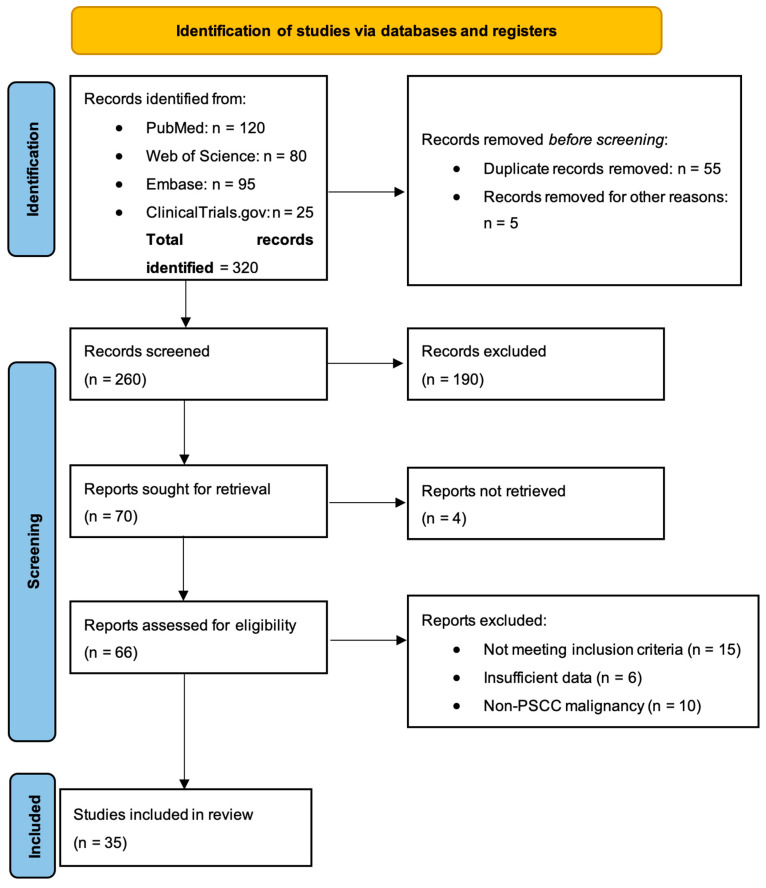
PRISMA 2020 flow diagram for new systematic reviews in the field of penile cancer (Appendix A).

**Table 1 cancers-17-00883-t001:** WHO classification of tumors of the penis [3].

Non-HPV-Related	HPV-Related
Squamous cell carcinoma (SCC), usual type	Basaloid SCC
Pseudohyperplastic carcinoma	Papillary basaloid carcinoma
Verrucous carcinoma	Warty carcinoma
Carcinoma cuniculatum	Warty-basaloid carcinoma
Papillary carcinoma NOS	Clear-cell carcinoma
Adenosquamous carcinoma	Lymphoepithelioma-like carcinoma
Sarcomatoid carcinoma	
Mixed squamous cell carcinoma	

**Table 2 cancers-17-00883-t002:** Overview: Clinical investigation of the immune checkpoint inhibitors in penile cancer in alphabetical order.

Substance Name	Brand Name	Substance Number	Pharmaceutical Company	Antibody Type	Target
Atezolizumab	Tecentriq	MPDL3280A	Genentech/Roche South San Francisco, CA, USA/ Basel, Switzerland	IgG1-humanized	PD-L1
Avelumab	Bavencio	MSB0010718C	Merck KGaA Darmstadt, Germany	IgG1-human	PD-L1
Cemiplimab	Libtayo	REGN2810	Regeneron Pharmaceuticals Tarrytown, NY, USA	IgG4-human	PD-1
Dostarlimab	Jemperli	GSK-4057190	GSK Brentford, UK	IgG4-human	PD-1
Ipilimumab	Yervoy	MDX-010	Bristol-Myers Squibb Lawrenceville, GA, USA	IgG1k-human	CTLA-4
Nivolumab	Opdivo	BMS-936558	Bristol-Myers Squibb Lawrenceville, GA, USA	IgG4-human	PD-1
Pembrolizumab	Keytruda	MK-3475	Merck & Co Rahway, NJ, USA	IgG4-humanized	PD-1
Retifanlimab	Zynyz	INCMGA00012	Incyte Wilmington, DE, USA	IgG4-human	PD-1

**Table 3 cancers-17-00883-t003:** (**A**) Comparative overview of the listed Phase 2 trials in advanced penile cancer with reported results. (**B**) Comparative overview of the listed Phase 2 trials in advanced penile cancer with pending results.

NCT Number (Trial Name)	Status	Intervention	Primary End Point	Number of Patients	Outcome
(**A**)					
**NCT03774901** **(PULSE)**	Recruiting	Avelumab maintenance therapy after disease control achieved after first-line platinum-based chemotherapy	PFS	32 patients planned, 9 patients evaluated for interim analysis	Survival without progression or death at 3 months 63.5% at 6 months 63.5% at 12 months 63.5% and 15 months 42.3% Overall survival at 12 months 88.9% and 15 months 88.9%
**NCT04224740** **(HERCULES)**	Completed	Pembrolizumab combined with cisplatin-based chemotherapy in first line	ORR	33 patients were eligible for efficacy analysis	ORR assessed by investigator 39.4% ORR central review 42.4% CBR assessed by investigator 45.5% mDOR 5.9 months mPFS 5.4 months mOS 9.6 months
**NCT03686332** **(PERICLES)**	Completed	Atezolizumab with or without radiotherapy for advanced PSCC	PFS at one year at least 35%	32 patients enrolled	PFS at one year: 12.5% ⟶ primary endpoint not met mOS. 11.3 months ORR: 16.7%
**NCT03012581** **(AcSé)**	Completed	Nivolumab monotherapy in chemotherapy-refractory PSCC	ORR	43 patients enrolled	12-week ORR: 14% mPFS 2.9 months mOS: 8.5 months 6-month PFS rate: 27.9% 6-month OS rate: 34.5%
**NCT04231981** **(ORPHEUS)**	Completed	Retifanlimab monotherapy in first-line PSCC	ORR	18 patients enrolled	ORR 16.7% mDOR 3.3 months CBR 22.2% mPFS 2.0 months mOS 7.2 months
(**B**)					
**NCT06353906** **(PRIAM)**	Recruiting	Induction therapy with carboplatin/paclitaxel + pembrolizumab for locoregionally advanced PSCC	pCR	50 patients planned	Not reported
**NCT05526989**	Recruiting	Niraparib + dostarlimab in relapsed/refractory penile cancer	ORR	25 patients planned	Not reported
**NCT03391479** **(ALPACA)**	Recruiting	Avelumab in PSCC patients unfit for or refractory to platinum-based chemotherapy	ORR	24 patients planned	Not reported
**NCT03357757 (LATENT)**	Active, not recruiting	Avelumab with valproic acid in virus-associated cancers, including p16-positive PSCC	ORR	39 patients in total	Not reported
**NCT03866382**	Recruiting	Nivolumab/ipilimumab + cabozantinib for rare genitourinary tumors, including penile cancer	ORR	314 patients in total	Not reported
**NCT04475016**	Completed	Triprilimab (anti-PD-1) with nimotuzumab (anti-EGFR) and the TIP chemotherapy regimen (paclitaxel, ifosfamide, cisplatin) in locally advanced PSCC	pCR	29 patients planned	Not reported

CBR, clinical benefit rate; mDOR, median duration of response; mOS, median overall survival; mPFS, median progression-free survival; ORR, overall response rate; pCR, pathological complete remission.

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
