# Peer review of "Immunotherapy in the Management of Penile Cancer—A Systematic Review"

_cancers, 2025, doi:10.3390/cancers17050883_

Round 1

Reviewer 1 Report

Comments and Suggestions for Authors

The manuscript lacked all the structure of a syst review.

Where is the M&M section with the saerch criteria? The PICOS protocol? The PRISMA flow diagram? The prospero registratrion? -> these fetures are foundamental parts of a syst review, and in their absence the work does not reach an adequate level of scientific accuracy.

The topic is interesting. The manuscript sounds more like  a narrative reivew,

The introduction is way too long

Author Response

Thank you for your thoughtful and constructive comments and remarks. Your valuable feedback has significantly contributed to enhancing the quality and clarity of our manuscript.

In the revised version, we have implemented the Methods section and the PRISMA chart flow.

Prospero registration: PROSPERO does not accept reviews that have been completed.

Reviewer 2 Report

Comments and Suggestions for Authors

Dear Authors,

I read with interest your paper, that is relevant. However, I would like to further clarify some points

1) methods: your paper should be a systematic review, however there are proper rules to follow, reported in the journal guidelines:

Systematic review articles present a detailed investigation of previous research on a given topic, and use clearly defined search parameters and methods to identify, categorize, analyze, and report aggregated evidence on a specific topic. A manuscript of this type should follow the PRISMA guidelines (https://www.mdpi.com/editorial_process#standards). Systematic reviews should include a Methods section. 

Please extensively report and state how you searched, selected and then reported the articles

Author Response

In the revised version, we have implemented the Methods section and the PRISMA chart flow.

We comprehensively explained how we searched for and selected the articles.

Reviewer 3 Report

Comments and Suggestions for Authors

The authors perform an exhaustive review of immunotherapy in penile cancer. This is an important topic that is clearly of interested to the field. The introduction (pages 2-3) covers the essential background information on penile cancer and the main immunotherapy agents (table 2). Then there is an overview of the most important trials in this space (pages 4-7, table 3), followed by the discussion and conclusion.  

Overall, this is a well rounded manuscript that can be considered for publication in the journal with some modifications:

The title suggests that this is a systematic review. In this case a protocol should be established explaining (a) how studies were selected for the review process (inclusion, exclusion criteria), (b) how the studies were found (which databases, which keywords were used etc), (c) how the data were synthesized, if applicable. The process should be documented according to guidelines such as the PRISMA statement (https://www.prisma-statement.org/). If the review was not performed in a systematic manner, it may alternatively be possible to indicate that this was a narrative (non-systematic) review.  

Page 2, lines 87-88: should provide precise ORR and mPFS/mOS for chemotherapy. E.g. TIP has shown an objective response rate of 50% in a neoadjuvant phase II trial (J Clin Oncol 2010;28:3851-3857). Although these responses are short-lived, it is not accurate that chemotherapy has “limited efficacy” in my opinion.

Page 3, lines 105-107: I think it is necessary to clarify whether PDL1 expression is prognostic or predictive of benefit from immunotherapy. Both can be true, but they are not equivalent. For example, PDL1 expression may be associated with bad prognosis irrespective of the treatment that is being delivered.

Page 4, lines 124-125: Please elaborate. What is exact % of patients who present durable responses? And what qualifies as a durable response? Do the authors specifically refer to the (rare) patients with MSI-high tumors?

Page 6, table 3: it may be useful to separate the trials who have reported results from those that are pending, for example table 3A and 3B. Given the small number of trials, and the small number of patients in each trial, I would also considering including phase I trials, such as ref 38, in the table in order to make it more exhaustive.

Finally, I would suggest separating the introduction (sections 1 and 1.1) from the main subject (sections 1.2 to 1.3) in the structure of the manuscript by using different headings and numberings. Simply put, sections 1.2 and 1.3 could become a section 2 “Results” or similar.

Author Response

Thank you for your detailed and constructive feedback. Your thoughtful remarks have helped us improve the manuscript, and we appreciate your careful review.

The title suggests that this is a systematic review. In this case a protocol should be established explaining (a) how studies were selected for the review process (inclusion, exclusion criteria), (b) how the studies were found (which databases, which keywords were used etc), (c) how the data were synthesized, if applicable. The process should be documented according to guidelines such as the PRISMA statement (https://www.prisma-statement.org/). If the review was not performed in a systematic manner, it may alternatively be possible to indicate that this was a narrative (non-systematic) review.

-Study Selection: Clear inclusion and exclusion criteria were defined to ensure the relevance and quality of the included studies.

-Search Strategy: A comprehensive literature search was conducted using databases such as PubMed, Embase, and Web of Science. Keywords and search terms were carefully selected to capture relevant studies on penile squamous cell carcinoma (PSCC), immune checkpoint inhibitors, and MSI-H status.

-Data Synthesis: The extracted data were systematically synthesized, with a focus on response rates, patient characteristics, and outcomes.

Page 2, lines 87-88: should provide precise ORR and mPFS/mOS for chemotherapy. E.g. TIP has shown an objective response rate of 50% in a neoadjuvant phase II trial (J Clin Oncol 2010;28:3851-3857). Although these responses are short-lived, it is not accurate that chemotherapy has “limited efficacy” in my opinion.

In the revised version we wrote:

Penile cancer is an uncommon malignancy, and in advanced stages, treatment op-tions are limited. Traditional platinum-based chemotherapies, such as the TIP regimen (paclitaxel, ifosfamide, and cisplatin), have demonstrated objective response rates of ap-proximately 50% in neoadjuvant settings, with a pathological complete response rate of 10%. However, these responses are often short-lived, underscoring the necessity for novel therapeutic approaches, including immune checkpoint inhibitors (ICIs).

In the revised version we wrote:

Programmed death-ligand 1 (PD-L1) expression is observed in up to 79% of both primary and metastatic penile squamous cell carcinomas (SCC), suggesting its potential as a predictive biomarker for the efficacy of immune checkpoint inhibitors (ICIs) [20-23]. Notably, PD-L1 expression has been associated with poor prognosis, particularly in human papillomavirus (HPV)-negative penile tumors, indicating that higher PD-L1 levels correlate with adverse clinical outcomes. This dual role of PD-L1—as a prognostic marker and as a predictor of response to ICIs—underscores its significance in identifying patient subgroups that may derive the greatest benefit from anti-PD-1/PD-L1 therapies. However, it is essential to recognize that while PD-L1 expression is associated with poor prognosis, it is predictive for the efficacy of immunotherapy [23-25].

Page 4, lines 124-125: Please elaborate. What is exact % of patients who present durable responses? And what qualifies as a durable response? Do the authors specifically refer to the (rare) patients with MSI-high tumors?

In the revised version we wrote:

Clinical studies have demonstrated durable responses in a small subset of patients with advanced penile cancer – particularly those with MSI-High status present in approximately 1% of all cases- who received ICIs after progressing on prior chemotherapy. Durable response refers to an objective response lasting at least 6 months.

Page 6, table 3: it may be useful to separate the trials who have reported results from those that are pending, for example table 3A and 3B. Given the small number of trials, and the small number of patients in each trial, I would also considering including phase I trials, such as ref 38, in the table in order to make it more exhaustive.

Finally, I would suggest separating the introduction (sections 1 and 1.1) from the main subject (sections 1.2 to 1.3) in the structure of the manuscript by using different headings and numberings. Simply put, sections 1.2 and 1.3 could become a section 2 “Results” or similar.

We implemented these recommendations.

Reviewer 4 Report

Comments and Suggestions for Authors

The review on immunotherapy in penile cancer (PeCa) is an interesting description of the mechanisms and a compilation of current and future approaches in the treatment of this cancer. 
Basically, the authors only focused on the results of the triali without going into the essence of the problem and without explaining the individual mechanisms.
Certain elements should be added/modified for the paper to be accepted for publication.
- There are no data on the differential survival of patients according to the underlying TNM.
- Does 5-fluorouracil or imiquimod or cis-platinum work?
- What is the rationale for evaluating p16 (or perhaps p16INK4A) in HPV-positive cancers? Does p16 remain intact in HPV-negative cancers?
- It is essential to provide a figure that explains the location and function of PD-1, PD-L1, CTLA-4 and other immune checkpoint proteins on the surface of tumour cells and TME cells.
- The mechanism of lymphocyte exhaustion is presented in 1 line, not even a sentence. More information is needed about this mechanism, how it works, what effect it has on the lymphocyte and on which TME cell populations? Is it only on CD3+8+ or does it also act on other cells?
- What is the mechanism of differential PD-L1 expression in HPV-negative PeCa?
- Why is ICI recommended in patients with high MSI and high TMB index? What is the basis for this?
- What is PARP?
- Because the authors write that 'El Zarif et al conducted an international, retrospective study of 92 patients with ad-214 vanced PSCC treated with immune checkpoint inhibitors (ICIs). The overall response rate was 13%. However, the use of existing ICIs may be more of a failure than a success.
- Since PD-L1 expression is much higher in HPV-negative cancers than in HPV-positive cancers, and since OS for HPV-positive cancers is better than for negative cancers, what is the justification for recommending anti-HPV vaccination in this cancer?

Author Response

We are grateful for your thoughtful and constructive comments. Your valuable suggestions have enhanced the clarity and rigor of our manuscript, and we appreciate your expertise and attention to detail.

- There are no data on the differential survival of patients according to the underlying TNM.

In the revised version we wrote:

PSCC prognosis varies significantly based on the TNM staging at diagnosis. The TNM system evaluates the primary tumor (T), regional lymph node involvement (N), and distant metastasis (M).

In localized stages (T1-T2, N0, M0), where the cancer is confined to the penis without lymph node involvement or distant spread, the 5-year relative survival rate is approximately 79%. As the disease progresses to regional spread (any T, N1-N2, M0), involving nearby lymph nodes but no distant metastasis, the 5-year relative survival rate decreases to around 51%. In advanced stages with distant metastasis (any T, any N, M1), where cancer has spread to other parts of the body, the 5-year relative survival rate drops significantly to approximately 9%.

Further stratification based on lymph node involvement provides additional insights. Patients with no regional lymph node metastasis (N0) have over a 90% 5-year survival rate. Those with metastasis in a single inguinal lymph node (N1) have nearly a 75% 5-year survival rate. Metastasis in multiple or bilateral inguinal lymph nodes (N2) corresponds to about a 60% 5-year survival rate. When metastasis involves pelvic lymph nodes or there is extranodal extension (N3), the 5-year survival rate is approximately 35%.

- Does 5-fluorouracil or imiquimod or cis-platinum work?

In the revised version we wrote

Topical therapy with imiquimod (IQ) or 5-fluorouracil (5-FU) are effective non-invasive first-line treatment options for the treatment of penile intraepithelial neoplasia (PeIN).

We implemented this section:
Penile cancer is an uncommon malignancy, and in advanced stages, treatment op-tions are limited. Chemotherapy regimens, such as traditional platinum-based chemo-therapies, such as the TIP regimen (paclitaxel, ifosfamide, and cisplatin), have demon-strated objective response rates of approximately 50% in neoadjuvant settings, with a pathological complete response rate of 10%.

- What is the rationale for evaluating p16 (or perhaps p16INK4A) in HPV-positive cancers?

Does p16 remain intact in HPV-negative cancers?

In the revised version we wrote

The pathogenesis of penile cancer is a multifactorial process influenced by various risk factors. Approximately one-third of all cases are associated with infection by the human papillomavirus (HPV). Immunohistochemical detection of the p16 protein serves as a surrogate marker for HPV-associated malignancies. In HPV-positive cancers, the viral oncoprotein E7 inactivates the retinoblastoma (Rb) protein, leading to upregulation of p16. This overexpression is commonly observed in HPV-associated malignancies, including PSCC. The most frequently identified HPV serotypes in penile cancer include HPV 16, 18, 31, 33, 45, 56, and 65 [10]

- It is essential to provide a figure that explains the location and function of PD-1, PD-L1, CTLA-4 and other immune checkpoint proteins on the surface of tumour cells and TME cells.

In the revised version we wrote

Thank you for your thoughtful suggestion regarding the inclusion of a figure illustrating the location and function of PD-1, PD-L1, CTLA-4, and other immune checkpoint proteins within the tumor microenvironment (TME). This is a systematic review, our primary focus is to synthesize and analyze the existing clinical evidence on immune checkpoint inhibitors in PSCC, rather than providing a mechanistic overview typically found in basic immunology texts. Including a figure depicting checkpoint protein locations and functions might shift the focus of the manuscript away from its clinical emphasis. However, we have clarified and expanded relevant sections to ensure that readers can easily understand the role of immune checkpoints within the context of PSCC without the need for additional visual representations.

- The mechanism of lymphocyte exhaustion is presented in 1 line, not even a sentence. More information is needed about this mechanism, how it works, what effect it has on the lymphocyte and on which TME cell populations? Is it only on CD3+8+ or does it also act on other cells?

In the revised version we wrote

In the context of penile squamous cell carcinoma (PSCC), the PD-1/PD-L1 axis plays a pivotal role in T cell exhaustion. PD-1, predominantly expressed on activated CD8+ T cells, binds to PD-L1 expressed on tumor cells and antigen-presenting cells within the TME, leading to T cell dysfunction. This interaction not only suppresses cytotoxicity but also limits T cell proliferation and cytokine release. While CD3+CD8+ T cells are particularly susceptible to this exhaustion pathway, emerging evidence suggests that CD4+ T cells and NK cells also exhibit PD-1-mediated functional impairments, contributing to immune evasion and tumor progression.

- What is the mechanism of differential PD-L1 expression in HPV-negative PeCa?

In the revised version we wrote

In penile squamous cell carcinoma (SCC), programmed death-ligand 1 (PD-L1) expression is more common in HPV-negative cases due to factors like higher tumor mutational burden (TMB), chronic inflammation, and immune microenvironment composition. Higher TMB produces neoantigens that trigger immune responses, prompting tumor cells to upregulate PD-L1. Chronic inflammation promotes continuous immune cell infiltration, sustaining PD-L1 expression. Additionally, CTLA4-positive immune cells in the tumor microenvironment further increase PD-L1 levels. In contrast, HPV-positive SCCs, with lower TMB and inflammation, show reduced PD-L1 expression, as their viral oncogenes elicit weaker immune responses

- Why is ICI recommended in patients with high MSI and high TMB index? What is the basis for this?

In the revised version we wrote

MSI-High and TMB-High are biomarkers associated with increased neoantigen formation, enhancing tumor immunogenicity. This heightened immunogenicity renders such tumors more susceptible to immune checkpoint blockade, as the immune system can better recognize and target cancer cells.

- What is PARP?

In the revised version we wrote

PARP is an enzyme family involved in DNA repair, particularly the base excision repair pathway. Inhibition of PARP leads to the accumulation of DNA damage, resulting in cell death, especially in cancer cells deficient in other DNA repair mechanisms (e.g., BRCA mutations).

- Because the authors write that 'El Zarif et al conducted an international, retrospective study of 92 patients with advanced PSCC treated with immune checkpoint inhibitors (ICIs). The overall response rate was 13%. However, the use of existing ICIs may be more of a failure than a success.
We rewrote this section:

El Zarif et al. performed an international retrospective study involving 92 patients with advanced PSCC who received treatment with immune checkpoint inhibitors (ICIs). The overall objective response was 13%, which was rather low.

- Since PD-L1 expression is much higher in HPV-negative cancers than in HPV-positive cancers, and since OS for HPV-positive cancers is better than for negative cancers, what is the justification for recommending anti-HPV vaccination in this cancer?

HPV infection is itself a well-known risk factor for the pathogenesis of various cancers, including PSCC. Thus, if the patient is not infected with HPV, the cancer would not occur in the first place.

Round 2

Reviewer 2 Report

Comments and Suggestions for Authors

Dear Authors,

thank you for extending the methods section and to complete the prisma check-list

Article is now fine

Kind regards